# Complete Meniscus Removal Method for Broadband Liquid Characterization in a Semi-Open Coaxial Test Cell [note 1]

**DOI:** 10.3390/s19092092

**Published:** 2019-05-06

**Authors:** Michał Kalisiak, Wojciech Wiatr

**Affiliations:** Institute of Electronic Systems, Warsaw University of Technology, 00-665 Warsaw, Poland; W.Wiatr@elka.pw.edu.pl

**Keywords:** complex permittivity, permeability, microwave measurements, liquids, meniscus, vector network analyzer, network de-embedding

## Abstract

We present a new technique for broadband liquid characterization using a semi-open, vertically oriented test cell that is measured with a calibrated vector network analyzer in three states: the empty one and filled with two different volumes of the liquid under test. Using the measurements, we de-embed a transfer matrix representing a volume increment of the liquid sample and determine its column height with a novel closed-form formula. Thanks to the de-embedding, the method enables one, for the first time, to completely remove effects caused by a reproducible meniscus forming the top surface of a liquid tested in the cell and determine not only the propagation constant, but also characteristic impedance of the liquid sample, from which its permittivity and permeability are calculated. The results are highly consistent, because all the measurements are performed without disassembling the cell. We validate experimental results of the meniscus removal method by comparing them with reference data and outcomes of the Nicolson–Ross–Weir method.

## 1. Introduction

The precise value of liquid complex permittivity and, less often, permeability is required in many fields of science, technology and industry such as chemistry, biology, medicine, agriculture, geophysics, radio communications, remote sensing, etc. Reference liquids are required for calibrating dielectric and microfluidic sensors [1] applied in dielectric spectroscopy [2]. At microwave frequencies, the most precise and reliable data are typically acquired by measuring liquid samples in test cells with vector network analyzers (VNAs) operating over very broadband frequency ranges. The permittivity is then determined from the VNA measurement data, but its accuracy still heavily depends on test cell design and methods applied to its extraction.

In the literature, there are diverse broadband methods for determining properties of liquids with a VNA [3,4,5,6]. Most of them have their roots in the idea of the transmission-reflection (T/R) technique, known also as the Nicolson-Ross-Weir (NRW) approach [7,8] which was introduced to measure properties of isotropic, homogeneous solid state materials. Typical measurements of liquids are carried out in a wave guiding cell whose space is closed by dielectric plugs. The liquid under test (LUT) has to fill this space completely, but it might be difficult due to air bubbles that need to be utterly removed before the measurement. Since the plugs are fixed, usability of such a structure is limited to specific liquids as regard their attenuation. Moreover, to correctly extract the LUT permittivity, the plugs affecting the measurements, have to be properly modeled, what, unfortunately, results in rather complex formulae [9,10].

Semi-open cells operating in vertical position, which are partly filled with liquid and air, as a waveguide one presented in [3] are more versatile. They allow dosing certain volume of a liquid to provide the best conditions for measuring the sample. Moreover, the actual height of the liquid column in the cell can be determined from the reflection coefficient measurements [3] and applied to the permittivity computation with the NRW formulae [8]. A similar approach to measuring samples in semi-open coaxial cells was presented in [11]. Both techniques ignored, however, the fact that the top surface of the liquid column may depart from a flat and transversal plane as it is usually assumed in the T/R methods. In reality, the surface is curved due to meniscus and this may cause errors in the determination of column height and consequently in the liquid permittivity characterization.

The issue of meniscus has been recently addressed in [12,13] where a more advanced technique than [11] has been introduced. Instead of two, this new technique exploits three measurement states, i.e., the empty-cell state and the other two with different volumes of liquid. Since all the measurements are performed without disassembling the cell, the measurement results exhibit higher consistency. The method utilizes the transfer matrices measured to form a similarity transformation, from which the propagation constant and the permittivity of liquid are extracted. However, the heights of liquid columns, necessary to this end, were still determined with the Somlo’s method [3], i.e., without considering the meniscus. A search [13] on meniscus effects in the permittivity measurements of distilled water samples, performed up to 18 GHz in a 7 mm line-standard test cell, showed that the errors caused by meniscus are rather small. Unfortunately, finite residual errors of the VNA calibration impeded discerning them in the de-embedded scattering parameters of the samples or in the column height determination. However, comparisons of the permittivity results calculated with different methods demonstrated evident advantages of this new three-state technique [12,13]. Nevertheless, one may expect that meniscus may have a stronger influence on accuracy of such permittivity measurements at higher frequencies when wave guiding structures of smaller size are required.

In this paper, we further improve the three-state technique with a novel algorithm for processing the transfer matrices measured. This algorithm de-embeds a transfer matrix representing the volume increment of the liquid sample and yields its column height, calculated using a novel closed-form formula. Therefore, the method is, for the first time, capable of completely removing any impact of reproducible meniscus on the permittivity and permeability determination. In Section 2, we present our model for the sample measurement and a novel algorithm for determining the height increment of the sample and its propagation constant along with the characteristic impedance. We validate the meniscus removal method in Section 3 by comparing our experimental results obtained for distilled water, propan-2-ol (IPA), and 50% aqueous solution of IPA with relevant results calculated using the NRW method [3,11], and with available reference data.

## 2. Theory

A sketch of a vertically oriented semi-open coaxial test fixture, employed for measuring liquids is shown in Figure 1 in its three different measurement states. As seen there, these states are related to three volumes of the LUT in the test cell; zero (empty cell), initial and final ones, which will be indexed with k=0,1,2, respectively. The mathematical description of this fixture and thus the theory of meniscus-removal method, we introduce here, is built on the underlying assumption about single mode propagation of TEM (transverse electromagnetic) waves in the test fixture. Then, the scattering matrices S of the fixture measured with a calibrated VNA are converted to the transfer ones according to
(1)$$T=\left[\begin{array}{cc} T_{11} & T_{12}\\ T_{21} & T_{22} \end{array}\right]=\frac{1}{S_{21}}\left[\begin{array}{cc} -\det S & S_{11}\\ -S_{22} & 1 \end{array}\right]\;,$$
because such a notation enables describing each state of the cell in a compact form.

For a *k*-th state, the transfer matrix of the fixture can be written down as:(2)$$T_{f}^{\left(k\right)}=T_{a}^{\left(k\right)}T_{s}^{\left(k\right)}T_{b}\;,$$
where $T_{a}^{\left(k\right)}$ and $T_{s}^{\left(k\right)}$ represent relevant sections of the cell filled with air and liquid sample, respectively, while $T_{b}$ describes the bottom part of the fixture consisting of the plug and the airline section at the port 2. All these matrices are referenced to the same characteristic impedance $Z_{c}$ of the airline the cell is based on.

In the case of empty cell, $T_{s}^{\left(0\right)}=I$, where I is the identity matrix and then (Equation 2) reduces to
(3)$$T_{f}^{\left(0\right)}=T_{a}^{\left(0\right)}T_{b}\;,$$
where, for given both the propagation constant $\gamma_{a}$ and the physical length $l_{a0}$ of the airline section, its matrix $T_{a}^{\left(0\right)}$ is determined by
(4)$$T_{a}^{\left(0\right)}=\left[\begin{array}{cc} e^{-\gamma_{a}l_{a0}} & 0\\ 0 & e^{\gamma_{a}l_{a0}} \end{array}\right]\;.$$

Thus, $T_{f}^{\left(0\right)}$ is utilized as in [11,12] to remove $T_{b}$ from (Equation 2) and extract components representing a *k*-th state of the test cell itself
(5)$$T_{c}^{\left(k\right)}=T_{a}^{\left(k\right)}T_{s}^{\left(k\right)}=T_{f}^{\left(k\right)}{\left(T_{f}^{\left(0\right)}\right)}^{-1}T_{a}^{\left(0\right)}\;.$$

On the assumption that the meniscus is reproducible in each state with liquid in the cell, the transfer matrices $T_{c}^{\left(k\right)}$ can be related to each other as follows:(6)$$T_{c}^{\left(1\right)}=T_{a}^{\left(1\right)}T_{s}^{\left(1\right)}=T_{a}^{\left(2\right)}T_{a\Delta }T_{s}^{\left(1\right)}\;,$$
(7)$$T_{c}^{\left(2\right)}=T_{a}^{\left(2\right)}T_{s}^{\left(2\right)}=T_{a}^{\left(2\right)}T_{s}^{\left(1\right)}T_{s\Delta }\;,$$
where $T_{s}^{\left(1\right)}$ stands for the initial LUT column of height $l_{s1}$, as shown in Figure 1b, while $T_{a\Delta }$ and $T_{s\Delta }$ represent the increment of the airline length and the LUT height, respectively
(8)$$\Delta l=l_{a1}-l_{a2}=l_{s2}-l_{s1}\;,$$
see Figure 1c. In contrast to both matrices in (Equation 6) and (Equation 7) related to the Δl increment it is assumed that, due to the meniscus, $T_{s}^{\left(1\right)}$ may generally represent electrical properties of an asymmetrical two-port network, for which sum of the off-diagonal terms may not be zero—see (Equation 1) for $S_{11}\neq S_{22}$.

To de-embed $T_{s\Delta }$, we need to remove the product of $T_{a}^{\left(2\right)}T_{s}^{\left(1\right)}$ from (Equation 7). Since $T_{a}^{\left(2\right)}T_{a\Delta }=T_{a\Delta }T_{a}^{\left(2\right)}$, we finally arrive at
(9)$$T_{s\Delta }={\left(T_{c}^{\left(1\right)}\right)}^{-1}T_{a\Delta }T_{c}^{\left(2\right)}\;.$$
where $T_{a\Delta }$ is described by (Equation 4) for Δl substituting $l_{a0}$. After multiplying the matrices in (Equation 9), we get
(10)$$T_{s\Delta }=\left[\begin{array}{cc} T_{22c}^{\left(1\right)}T_{11c}^{\left(2\right)}e^{-\gamma_{a}\Delta l}-T_{12c}^{\left(1\right)}T_{21c}^{\left(2\right)}e^{\gamma_{a}\Delta l} & T_{22c}^{\left(1\right)}T_{12c}^{\left(2\right)}e^{-\gamma_{a}\Delta l}-T_{12c}^{\left(1\right)}T_{22c}^{\left(2\right)}e^{\gamma_{a}\Delta l}\\ T_{11c}^{\left(1\right)}T_{21c}^{\left(2\right)}e^{\gamma_{a}\Delta l}-T_{21c}^{\left(1\right)}T_{11c}^{\left(2\right)}e^{-\gamma_{a}\Delta l} & T_{11c}^{\left(1\right)}T_{22c}^{\left(2\right)}e^{\gamma_{a}\Delta l}-T_{21c}^{\left(1\right)}T_{12c}^{\left(2\right)}e^{-\gamma_{a}\Delta l} \end{array}\right]\;,$$
where quantities $T_{mnc}^{\left(k\right)}$ are the relevant terms of matrices $T_{c}^{\left(k\right)}$ for m,n=1,2. In (Equation 10), Δl is the sole unknown parameter. So we determine it from the condition of electrical symmetry regarding $T_{s\Delta }$ representing a section of the liquid sample between two ideal planes that are transversal to its axis. For such a sample $T_{12}=-T_{21}$ and then
(11)$$e^{2\gamma_{a}\Delta l}=\frac{T_{22c}^{\left(1\right)}T_{12c}^{\left(2\right)}-T_{21c}^{\left(1\right)}T_{11c}^{\left(2\right)}}{T_{12c}^{\left(1\right)}T_{22c}^{\left(2\right)}-T_{11c}^{\left(1\right)}T_{21c}^{\left(2\right)}}=r\;,$$
where *r* represents the above ratio of the measured quantities. This ratio can be expressed in terms of the scattering matrix by using the relationships (Equation 1). Then, it can be interpreted as a quotient of two reflection coefficients $\Gamma^{\left(1\right)}$ and $\Gamma^{\left(2\right)}$
(12)$$r=\frac{\Gamma^{\left(2\right)}}{\Gamma^{\left(1\right)}}=e^{2\gamma_{a}\Delta l}\;,$$
where
(13)$$\Gamma^{\left(k\right)}=S_{11c}^{\left(k\right)}+\frac{S_{12c}^{\left(k\right)}S_{21c}^{\left(k\right)}S_{22c}^{\left(i\right)}}{1-S_{22c}^{\left(k\right)}S_{22c}^{\left(i\right)}}\;,$$
while index i=1,2 for i≠k. Finally, we determine the length increment from (Equation 12)
(14)$$\Delta l=\frac{1}{2\gamma_{a}}\left(\ln\Gamma^{\left(2\right)}-\ln\Gamma^{\left(1\right)}\right)\;.$$

The above method and formulae for determining $T_{s\Delta }$ and Δl have been presented for the first time. For the de-embedding applied to sample’s volume increment, the method is capable of removing errors caused by the meniscus provided it is reproducible. Because the terms $T_{12}$ and $T_{21}$ may be exposed to errors in the reflection coefficient measurements performed by a VNA over a broad frequency range, we employ a robust statistical optimization to assess single value of Δl from the data calculated using (Equation 14) at each frequency.

Once Δl is assessed, the matrix $T_{s\Delta }$ can be directly calculated from (Equation 9). However due to unavoidable errors, we obtain a perturbated matrix ${\tilde{T}}_{s\Delta }$ that does not perfectly match its theoretical model:(15)$$T_{s\Delta }=T_{t}\left[\begin{array}{cc} e^{-\gamma_{s}\Delta l} & 0\\ 0 & e^{\gamma_{s}\Delta l} \end{array}\right]T_{t}^{-1}\;.$$
where $T_{t}$ denotes the transfer matrix of an abrupt transition on the boundary between air and the liquid
(16)$$T_{t}=\frac{1}{\sqrt{1-\Gamma_{s}^{2}}}\left[\begin{array}{cc} 1 & \Gamma_{s}\\ \Gamma_{s} & 1 \end{array}\right]\;,$$
while $\Gamma_{s}$ stands for the reflection coefficient at this interface
(17)$$\Gamma_{s}=\frac{Z_{s}-Z_{c}}{Z_{s}+Z_{c}}\;.$$

Since (Equation 15) represents a matrix similarity transformation, the trace of each side in (Equation 9) is the same and hence we get
(18)$$\mathrm{tr}\;T_{s\Delta }=e^{-\gamma_{s}\Delta l}+e^{\gamma_{s}\Delta l}=2\cosh\gamma_{s}\Delta l=\mathrm{tr}\;{\tilde{T}}_{s\Delta }\;.$$

From this, we determine the LUT propagation constant
(19)$$\gamma_{s}=\frac{1}{\Delta l}\mathrm{arcosh}\left(\frac{1}{2}\mathrm{tr}\;{\tilde{T}}_{s\Delta }\right)\;.$$

Having the eigenvalues of (Equation 15) determined, we extract its relevant eigenvectors, from which we then calculate the reflection coefficient:(20)$$\Gamma_{s}=\frac{-T_{21}}{T_{22}-e^{-\gamma_{s}\Delta l}}\;,$$
where $T_{22}$ and $T_{21}$ are elements of $T_{s\Delta }$. This leads us to characteristic impedance of the liquid-sample line section:(21)$$Z_{s}=Z_{c}\frac{1+\Gamma_{s}}{1-\Gamma_{s}}\;.$$

The determined quantities $\gamma_{s}$ and $Z_{s}$ allow us to extract both the relative permittivity $\varepsilon_{r}$ and permeability $\mu_{r}$ of LUT in the usual way—see (Equation 22), (Equation 23) in Section 3.

## 3. Experimental Results

The design of our coaxial test fixture for the liquid characterization followed the general idea of [3]. The fixture was based on 7 mm line standard with 7 mm laboratory precision connectors (LPC-7) [14]. For VNA calibration we used manufactured earlier airline standards [15]. To provide the same properties as those standards and keep costs low, body of the fixture was machined in an in-house workshop from a brass rod. Its body was equipped with a center conductor, a PTFE annular plug, as illustrated in Figure 1, and a stub pipe for dosing liquids through a small hole in the body. A picture of the fixture, vertically mounted in the measurement setup, is shown in Figure 2.

Prior to the characterization of liquids, we calibrated our VNA in the frequency range from 0.1 to 18 GHz with the airline standards [15] and the multiline method [16]. This calibration yielded, as its byproduct, such characteristics of the airline as its attenuation and phase constants as well as the characteristic impedance $Z_{s}$. These data were then utilized to calculate the permittivity and permeability of LUTs.

We measured scattering matrices of the fixture in all its three states for distilled water, IPA and 50% aqueous solution of IPA. We applied the meniscus removal technique, as described in the Section 2, to determine the propagation factor $\gamma_{s}$ and calculate the reflection coefficient $\Gamma_{s}$. Since the symmetry assumption on the de-embedded volume increment of the sample is of key importance for our approach, we first checked the ratio *r* (Equation 12) to see how well it adheres to the theory. In Figure 3 we show the magnitude and angle of this ratio versus frequency for all the liquids measured. Since for low-loss airline the magnitude of the left side (Equation 11) is close to unity, deviations from it seen in Figure 3a, may be attributed to some residual errors of the VNA calibration.

From the ratio (Equation 12), we calculated then the height increment Δl at each measurement frequency using (Equation 14). The results are showed in Figure 4 on relevant graphs for a different increment of each liquid with blue lines. Again, due to the residual errors, the curves evidence regular ripples that rise at low frequency end, because of increasing sensitivity of this calculation to errors, which results from a finite angular resolution of the characteristics shown in Figure 3b. To preclude effects of the abnormal deviations, we applied a robust statistical optimization for assessing a single value of Δl for (Equation 15). The optimized values of the sample height increments are shown with red dashed lines on the graphs along with their ±2% boundaries drawn with gray dots for better perceiving. These graphs evidence that the relative errors caused by the ripples decrease with the increment of column height that is, however, limited by liquid attenuation. So for the increment, the lowest relative errors can be attributed to LUT having the largest inclination of its characteristic in Figure 3b.

For the known height increments, we then determined the relative permittivity of liquids at the assumption of their relative permeability $\mu_{r}=1$. To this end, we utilized the propagation constant determined with (Equation 19), from which we calculated
(22)$$\varepsilon_{r}=\varepsilon_{r}^{\prime }-j\varepsilon_{r}^{\prime \prime }=-{\left(\frac{v_{a}\gamma_{s}}{\omega }\right)}^{2}\;,$$
where ω=2πf is the angular frequency and $v_{a}$ stands for the speed of light in air. We compared results of our new method with the outcomes of classic NRW method [11]. As this method employs only a single volume of liquid, we can determine the permittivity for the initial and the final volumes of LUT, independently. To calculate the height of liquid columns $l_{sk}$, we used solution proposed in [3]. Although for $\mu_{r}=1$, the NRW method allows calculating $\varepsilon_{r}$ from both the transmission and reflection coefficients independently [11], we decided to present here results of the former ones for their higher consistency.

The results of water permittivity measurements are shown in Figure 5. As a reference we used the data taken from [17], where authors proposed a refined model of water permittivity up to 25 THz for temperature range 0–100 °C. We can observe very good consistency of our results (red dashed-dotted lines) with the reference (blue lines). Relative difference in the real part of permittivity between our method and the reference does not exceed 2.5% in the entire bandwidth, except the first point at 0.1 GHz, where it reaches 6%. The imaginary part yields convergence better than 3.5% above 3.5 GHz. The NRW results for the initial volume of liquid (purple dashed lines) are rather distant from the reference. A better agreement is obtained for the final volume (green dots), because of its higher column of water, the impact of liquid surface distortions is smaller and thus results in improved accuracy.

The IPA permittivity results are presented in Figure 6 along with the reference data are taken from [18] at 30 °C (blue lines), but certified below 5 GHz. The measured characteristics have similar shape to the reference ones, although are shifted in frequency. The reason may be a high sensitivity of the IPA relaxation frequency (related to the peak of $\varepsilon_{r}^{\prime \prime }$ in Figure 6b) to the temperature [18]. Except the low frequency end, the NRW method used for the final volume of IPA (green dots) yields almost the same results as our meniscus removal method (red dashed-dotted lines), while outcomes for its initial volume (purple dashed lines) are apart of both for the same reason as stated before.

For the 50% aqueous solution of IPA, we, unfortunately, have not got relevant data for comparison except just static permittivity at room temperature [19] showed with blue crosses in Figure 7, confirming that values at low frequency end are reasonable. Likewise for clean IPA, our method (red dashed-dotted lines) yields similar results as NRW one for the final volume of liquid (green dots), whilst outcomes for its initial level (purple dashed lines) are not consistent with them.

Since our novel method allows us to calculate both $\gamma_{s}$ and $Z_{s}$, we are able to characterize magnetic liquids and determine their relative permittivity and permeability:(23)$$\varepsilon_{r}=-j\frac{v_{a}\gamma_{s}}{\omega }\;\frac{Z_{c}}{Z_{s}}\;,\;\mu_{r}=\mu_{r}^{\prime }-j\mu_{r}^{\prime \prime }=-j\frac{v_{a}\gamma_{s}}{\omega }\;\frac{Z_{s}}{Z_{c}}\;.$$

As an example of our generic approach, we show the permittivity and permeability of already presented, nonmagnetic IPA, but calculated without assuming any knowledge of its permeability. The results of permittivity obtained for IPA, shown in Figure 8a,b, agree well with all the previously shown in Figure 6 which were obtained at the assumption $\mu_{r}=1$. Agreement with that assumption is illustrated on the the permeability graphs shown in Figure 8c,d. Departures from it, easily discerned on the graphs, evidence an asset of the meniscus removal method (red dashed-dotted lines), whose results are close to the reference, however, with some perturbations at higher frequencies caused perhaps by residual errors of the VNA calibration.

## 4. Conclusions

We have presented a new technique for broadband liquid characterization using a semi-open, vertically oriented test cell that was measured with a calibrated VNA in three states: the empty-cell state and filled with the initial and the final volumes of the liquid. With the transfer matrices describing the measurements, we developed a novel method for de-embedding a volume increment of the liquid sample and derived a unique closed-form formula for exact determining its height increment from the electrical parameters measured by the VNA. With this method, we were able, for the first time, to completely remove the errors caused by the reproducible meniscus forming the top surface of the liquid tested and thus determine its permittivity and permeability with higher accuracy.

We verified this new method experimentally by measuring distilled water, IPA, and 50% aqueous solution of IPA. We compared the results with the reference data and relevant outcomes calculated with the NRW method as in [3,11]. In general, the meniscus removal method provides more precise results in a broader frequency range when comparing it to the classic NRW method. Since accuracy of the length and permittivity determination with this novel technique may be distorted by residual errors of the VNA calibration, we are going to investigate this issue further.

## Figures and Tables

**Figure 1 sensors-19-02092-f001:**
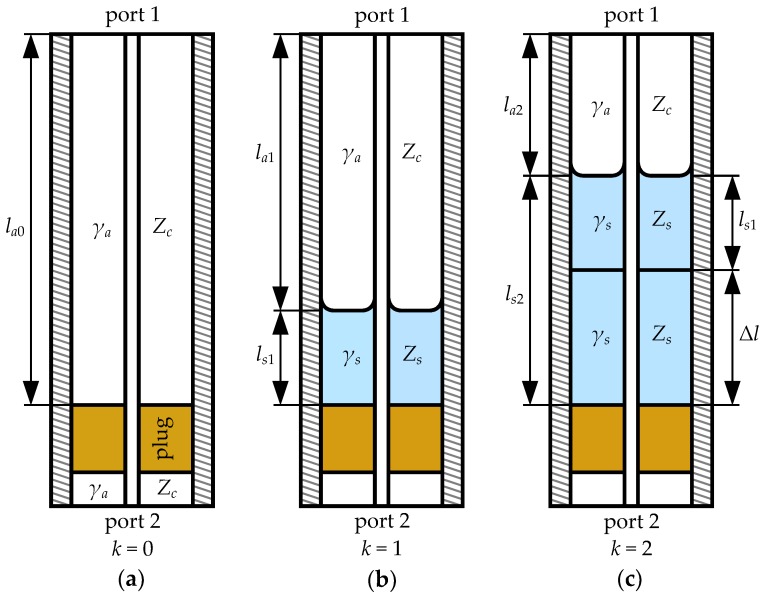
Sketch of the semi-open coaxial fixture in its three measurement states of liquid under test: (**a**) empty cell; (**b**) initial volume; (**c**) final volume. List of symbols: $l_{ak}$—height of *k*-th airline section (k=0,1,2); $l_{sk}$—height of *k*-th sample; Δl—height increment; $Z_{c}$, $Z_{s}$—characteristic impedances and $\gamma_{a}$, $\gamma_{s}$—propagation constants for the air and sample, respectively.

**Figure 2 sensors-19-02092-f002:**
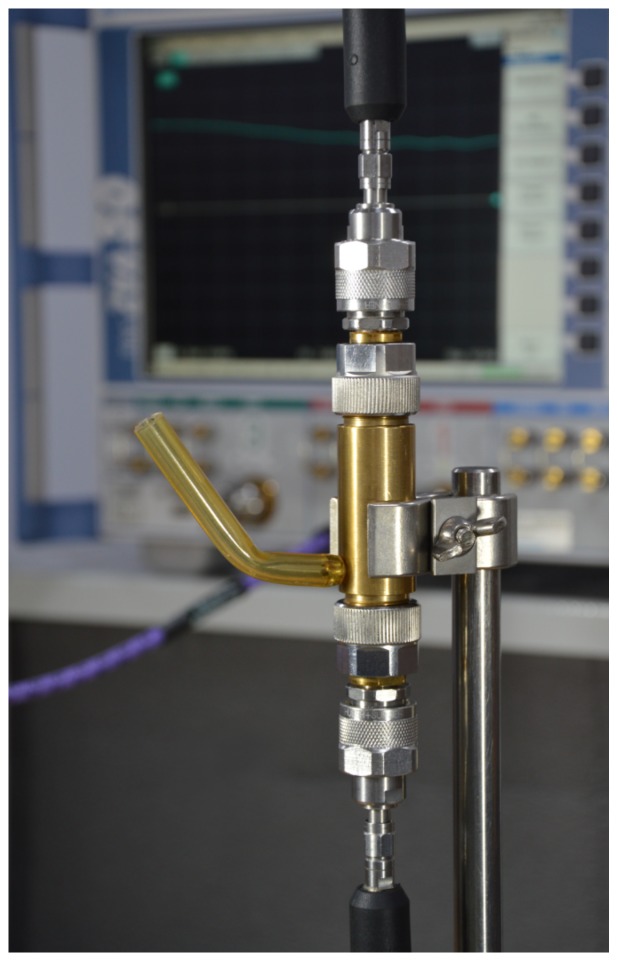
Measure setup.

**Figure 3 sensors-19-02092-f003:**
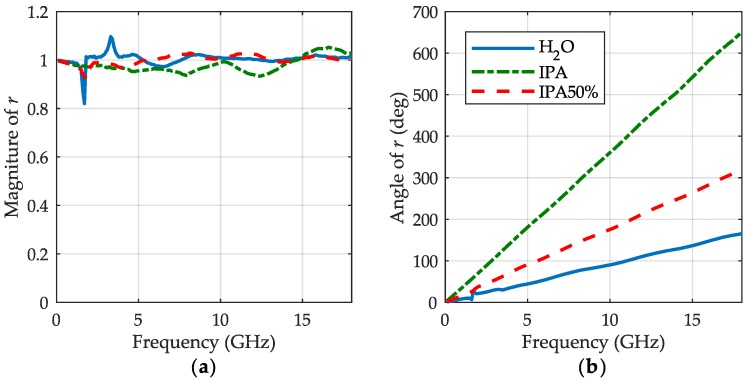
The magnitude (**a**) and the angle (**b**) of ratio *r* for distilled water (blue lines), IPA (green dashed-dotted lines), 50% aqueous solution of IPA (red dashed lines).

**Figure 4 sensors-19-02092-f004:**
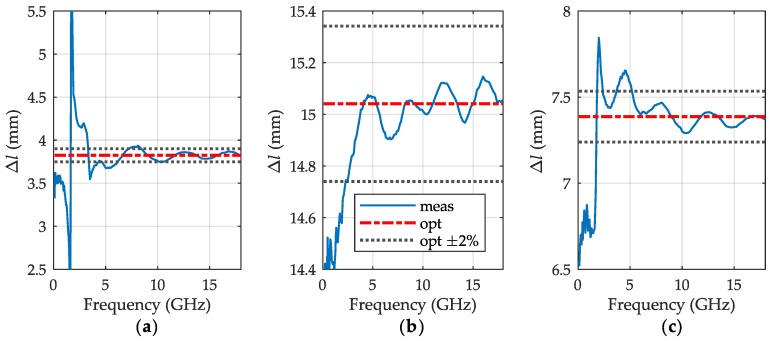
Height Δl of additional portion of liquid for (**a**) distilled water, (**b**) IPA and (**c**) 50% aqueous solution of IPA calculated with (Equation 14) (blue line) and its optimal value (red dashed-dotted line) with ±2% boundaries (gray dots).

**Figure 5 sensors-19-02092-f005:**
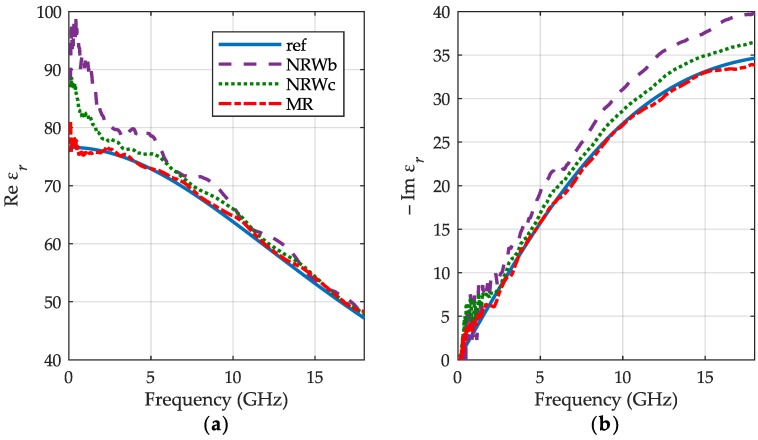
The relative permittivity of distilled water at 30 °C: (**a**) the real part $\varepsilon_{r}^{\prime }$; (**b**) the imaginary part $\varepsilon_{r}^{\prime \prime }$ obtained for the meniscus removal method (MR)—red dashed-dotted lines, the NRW method for the initial volume of water—purple dashed lines and its final volume—green dots with reference curves—blue lines.

**Figure 6 sensors-19-02092-f006:**
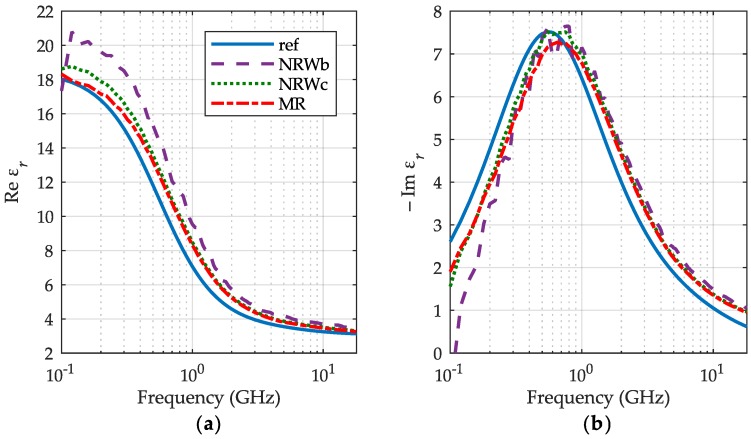
The relative permittivity of IPA at 30 °C: (**a**) the real part $\varepsilon_{r}^{\prime }$; (**b**) the imaginary part $\varepsilon_{r}^{\prime \prime }$ obtained for the meniscus removal method (MR)—red dashed-dotted lines, the NRW method for the initial volume of IPA—purple dashed lines and its final volume—green dots with reference curves—blue lines.

**Figure 7 sensors-19-02092-f007:**
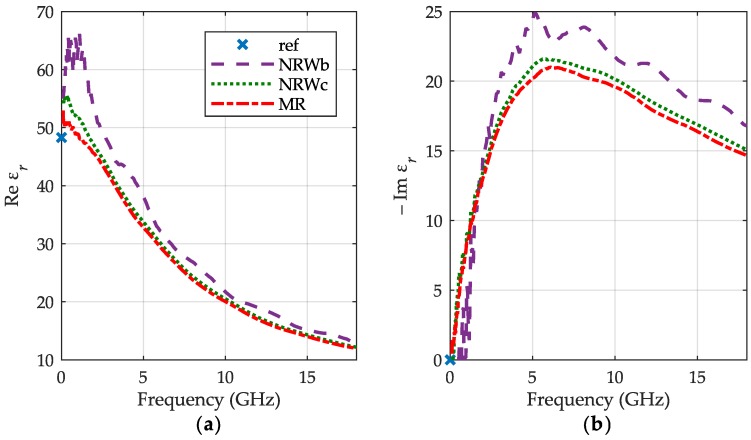
The relative permittivity of 50% aqueous solution of IPA at 30 °C: (**a**) the real part $\varepsilon_{r}^{\prime }$; (**b**) the imaginary part $\varepsilon_{r}^{\prime \prime }$ obtained for the meniscus removal method (MR)—red dashed-dotted lines, the NRW method for the initial volume of IPA—purple dashed lines and its final volume—green dots with reference point of static permittivity at room temperature—blue crosses.

**Figure 8 sensors-19-02092-f008:**
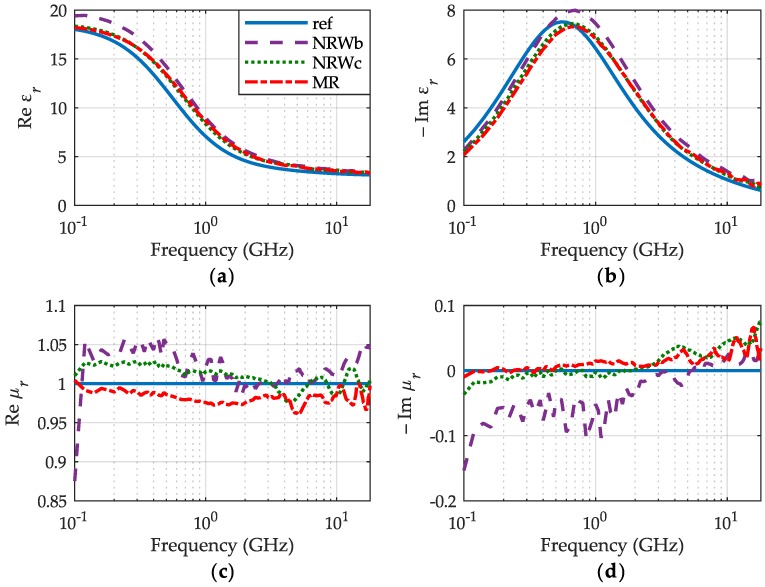
The relative permittivity (**a**) $\varepsilon_{r}^{\prime }$, (**b**) $\varepsilon_{r}^{\prime \prime }$ and permeability (**c**) $\mu_{r}^{\prime }$, (**d**) $\mu_{r}^{\prime \prime }$ of IPA at 30 °C, obtained for the meniscus removal method (MR)—red dashed-dotted lines, the NRW method for the initial volume of IPA—purple dashed lines and its final volume—green dots with reference curves—blue lines.

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
