# Peer review of "Complete Meniscus Removal Method for Broadband Liquid Characterization in a Semi-Open Coaxial Test Cell†"

_sensors, 2019, doi:10.3390/s19092092_

Round 1
Reviewer 1 Report
In the manuscript #486252 the authors describe a broadband microwave Transmission-Reflection (T/R) technique to determine permittivity of liquid materials in the frequency range from 0.1 GHz up to 18 GHz.
The conventional T/R methods, such as NRW, are suited for solid specimens with fixed dimensions enabling calibration at well-defined calibration planes. In the case of liquid specimens it is difficult to match the calibration plane positions where the calibration standards are attached, with the a actual wave propagation length in the specimen. Consequently the permittivity measurements of liquid materials are typically restrained to one port reflection from semi-infinite samples.
To extend the conventional T/R methodology into accurate two port T/R for liquids, the authors show that the selected propagation, length delta_L, can be determined from two measurement of the reflection coefficient (equation 11-13). Then the complex scattering parameters determined at the calibration planes can be transferred to the delt_l boundaries by solving the transfer matrix (1) with subsequent steps shown on pages 3-5. The presented permittivity results are reasonable and since the method is broadband it allows for more precise measurement and identification of, for example, contaminants through permittivity and characteristic relaxation time. In addition, the methodology appears to be capable of extracting magnetic permeability from the same measurement.
In my opinion, the contribution from the authors to the measurement science of dielectric liquid materials is significant and I recommend it for publication.
Some additional comments aimed at improving the manuscript readability:
1. Since there are only three measurements illustrated inFig.1 (a, b, c), I recommend that the math subscripts be labelled thru-out the manuscript accordingly. Subscript referring to, for example, k-states (eq (2) (5) ….(12), that have a general meaning and are not directly referring to the actually presented methodology, makes it difficult to follow the authors point of view.
2. More accurate link to Ref 14, where the authors reference their custom LPC7 connectors is needed. I was finally able to get a copy of that paper from https://ieeexplore.ieee.org/stamp/stamp.jsp?tp=&arnumber=4630155.
Nevertheless, I would recommend to change this LPC7 to the standard connector APC7 or describe the properties of the LPC7 in more detail if these are critical for the method.
3. A numerical example illustrating a step-by-step algorithm how to practically use the presented method, for example for permittivity of water at 10 GHz, would greatly increase the impact of the paper. Such example would be especially useful to practitioners that are not immediately familiar with mathematical handling of complex matrices.
Author Response
We would like to thank you for the positive reception of our paper and your review that has helped us to improve the paper.
The changes resulting from reviewer’s 1 comments are highlighted in yellow in uploaded “paper19_differences.pdf” file.
The changes resulting from reviewer’s 2 comments are highlighted in green in uploaded “paper19_differences.pdf” file.
Another changes are highlighted in cyan.
Point 1: Since there are only three measurements illustrated inFig.1 (a, b, c), I recommend that the math subscripts be labelled thru-out the manuscript accordingly. Subscript referring to, for example, k-states (eq (2) (5) ….(12), that have a general meaning and are not directly referring to the actually presented methodology, makes it difficult to follow the authors point of view.
Response 1: We agree that matrix indices were not consistent throughout the article. We have unified the naming system. Now it looks like this: Tx(k), where x says about section / medium / etc., and k=0,1,2 describes the number of measurement. We have also add information in Figure 1 which measurement corresponds to which k number.
Point 2: More accurate link to Ref 14, where the authors reference their custom LPC7 connectors is needed. I was finally able to get a copy of that paper from https://ieeexplore.ieee.org/stamp/stamp.jsp?tp=&arnumber=4630155.
Nevertheless, I would recommend to change this LPC7 to the standard connector APC7 or describe the properties of the LPC7 in more detail if these are critical for the method.
Response 2: We have added hyperlink to the reference you mentioned.
We have added a new reference to a paper “Guidance on using Precision Coaxial Connectors in Measurement” describing properties of LPC-7 connector that we used. The type of connector used is not critical for the method.
Point 3: A numerical example illustrating a step-by-step algorithm how to practically use the presented method, for example for permittivity of water at 10 GHz, would greatly increase the impact of the paper. Such example would be especially useful to practitioners that are not immediately familiar with mathematical handling of complex matrices.
Response 3: Thank you for that comment. Unfortunately our method cannot work on single frequency (especially on the higher one), as phase continuity of complex values has to be ensured. The whole method is explained in details in Section 2. “Theory” and allows for implementation, however, requires handling with complex matrices.
Thank you again for your review and the comments that have helped us to improve the paper.
Reviewer 2 Report
This paper, well written, provides a new approach to compensating for the meniscus when determining the dielectric properties of liquids in a transmission line test cell.
I have only a few suggested changes:
Line 23 diverse?
Line 28 'but' instead of 'what' ?
line 198 It would be helpful to speculate just a little on the likely cause of the error - do you consider it likely to be a part of errors in the VNA that have not been calibrated out, or test cell errors, perhaps incomplete compensation around the meniscus edges due to reflections? And are these errors included in those you wish to research further?
Author Response
We would like to thank you for the positive reception of our paper and your review that has helped us to improve the paper.
The changes resulting from reviewer’s 1 comments are highlighted in yellow in uploaded “paper19_differences.pdf” file.
The changes resulting from reviewer’s 2 comments are highlighted in green in uploaded “paper19_differences.pdf” file.
Another changes are highlighted in cyan.
Point 1: I have only a few suggested changes:
Line 23 diverse?
Response 1: Yes, thank you.
Point 2: Line 28 'but' instead of 'what' ?
Response 2: We have changed according to your proposal.
Point 3: line 198 It would be helpful to speculate just a little on the likely cause of the error - do you consider it likely to be a part of errors in the VNA that have not been calibrated out, or test cell errors, perhaps incomplete compensation around the meniscus edges due to reflections? And are these errors included in those you wish to research further?
Response 3: We agree this statement was ambiguous. We have stressed that the reason of the errors is probably imperfect calibration of vector network analyser. Yes, those are the errors we write in “Conclusions” we wish to investigate further.
Thank you again for your review and the comments that have helped us to improve the paper.
This manuscript is a resubmission of an earlier submission. The following is a list of the peer review reports and author responses from that submission.